# Performance Optimization for Phase-Sensitive OTDR Sensing System Based on Multi-Spatial Resolution Analysis

**DOI:** 10.3390/s19010083

**Published:** 2018-12-27

**Authors:** Yuanyuan Shan, Wenbin Ji, Qing Wang, Lu Cao, Feng Wang, Yixin Zhang, Xuping Zhang

**Affiliations:** 1The Key Laboratory of Intelligent Optical Sensing and Manipulation, Ministry of Education, Nanjing 210093, China; dg1523006@smail.nju.edu.cn (Y.S.); jiwb@nju.edu.cn (W.J.); mg1634003@smail.nju.edu.cn (Q.W.); mg1634013@smail.nju.edu.cn (L.C.); wangfeng@nju.edu.cn (F.W.); 2Institute of Optical Communication Engineering, Nanjing University, Nanjing 210093, China; 3The Key Laboratory of Modern Acoustics, Nanjing University, Nanjing 210093, China

**Keywords:** phase-sensitive optical time domain reflectometry (Φ-OTDR), multi-spatial resolution (MSR) analysis, signal-to-noise ratio (SNR)

## Abstract

This paper proposes and demonstrates a phase-sensitive optical time domain reflectometry (Φ-OTDR) sensing system with multi-spatial resolution (MSR) analysis property. With both theoretical analysis and an experiment, the qualitative relationship between spatial resolution (SR), signal-to-noise ratio (SNR) and the length of the vibration region has been revealed, which indicates that choosing a suitable SR to analyze the vibration event can effectively enhance the SNR of a sensing system. The proposed MSR sensing scheme offers a promising solution for the performance optimization of Φ-OTDR sensing systems, which can restore vibration events of different disturbance range with optimum SNR in merely a single measurement while maintaining the same detectable frequency range.

## 1. Introduction

Phase-sensitive optical time domain reflectometry (Φ-OTDR) has attracted growing interest as a practical and effective technology for applications of vibration detection, such as the monitoring of underwater acoustic and seismic signals [1], owing to its high sensitivity, fast response and multi-point measurement capability [2]. For an Φ-OTDR sensing system, both spatial resolution (SR) and signal-to-noise ratio (SNR) are critical indexes to evaluate the sensing performance. Depending on the signal of interest, Φ-OTDR sensing systems can be categorized into two main kinds: amplitude-discriminating Φ-OTDR and phase-discriminating Φ-OTDR [3].

In an amplitude-discriminating Φ-OTDR sensing system, the vibration event applied on the sensing fiber can be located by demodulating the amplitude variation of Rayleigh backscattering (RBS) light. For a given sensing fiber, the SNR of an amplitude-discriminating Φ-OTDR system can be considered as a random process of both time and distance along the fiber. To enhance the SNR, more energetic probe pulse should be used, which means a larger peak power of probe pulse or wider pulse width is required. However, the peak power of probe pulse is restricted by the optical nonlinear effect, resulting in a limited tuning range of the overall pulse energy. Instead, wider pulse is usually adopted to increase the pulse energy. Since the SR of Φ-OTDR is mainly determined by the probe pulse width, there will be a tradeoff between SR and SNR. The SNR can be enhanced by introducing more components or modules to the sensing system, such as distributed amplification [4,5] and ultra-weak fiber Bragg grating [6]. Such modifications apparently increase the overall complexity of the system. Digital signal processing (DSP) is another powerful tool that can effectively reduce the noise level without changing the sensing system structure, such as moving average and moving differential methods [7]. Subsequently, the edge detection and two-dimensional processing of RBS traces have been reported to increase the SNR [8]. Other than these time-domain signal processing methods mentioned above, the wavelet denoising method has been adopted to reduce the random noises induced by varied polarization states in different positions of the fiber [9]. A one-dimensional Fourier transform of RBS traces has been demonstrated to obtain the location and frequency information of vibration signals [10]. Adaptive matched filter was introduced in coherent detection-based Φ-OTDR systems to enhance SNR without using an additional optical amplifier [11].

Unlike amplitude-discriminating Φ-OTDR, the phase variation of the RBS is demodulated to restore the external vibration event for phase-discriminating Φ-OTDR, which could dramatically increase the sensitivity and fidelity of the sensing system during the reconstruction of vibration event [12]. However, the performance of phase-discriminating Φ-OTDR will be affected by the background noise, of which the level is commonly much higher than the system noise for in-field applications such as the structure health monitoring (SHM) of expressways, railways, bridges and dams. These structures themselves as well as the sensing fiber cables would be continuously disturbed by vehicles, trains, boats and tides that generating high background vibration noise. Therefore, many advanced DSP techniques to improve the sensing performance of phase-discriminating Φ-OTDR have been reported in literature. Frequency-division-multiplexing and matched filter algorithms to extend the frequency response range of Φ-OTDR has been proposed [13]. An adaptive 2D bilateral filtering algorithm has been adopted to enhance the SNR and extract weak vibration in an Φ-OTDR sensing system [14]. A denoising method based on empirical mode decomposition has been proposed to improve the SNR of Φ-OTDR [15]. However, the relationship between SNR, SR and the length of the vibration region has not been completely discussed yet. It is worth noting that even if the relationship is quantified, it is still difficult to achieve the optimal measurement by matching the SR with the length of the vibration event, since the length of vibration exerted on the sensing fiber cannot be known in advance.

In this paper, a model describing the qualitative relationship between SR, SNR and the length of the vibration region will be given and verified by experiments. Subsequently, a novel sensing scheme based on tri-pulses frequency division multiplexing (FDM) will be presented, which can perform multi-spatial resolution (MSR) analysis in a single measurement. With the proposed scheme, the SR and the length of vibration region can be matched in a more accurate way, and the performance optimization of a Φ-OTDR sensing system can be obtained without reducing the detection frequency range.

## 2. The Relationship between SR, SNR and the Length of Vibration Region

In current research on phase-discriminating Φ-OTDR, the vibration is commonly considered as exerting on a small region of the SR cell. The RBS lights from upstream and downstream of the vibration location are unaffected. By measuring the phase difference between the upstream and downstream location induced by vibration, the vibration waveform can be fully restored [16]. However, in practice, the vibration may be applied to a long section of the sensing fiber that contains many SR cells. These reference regions might be modulated by vibration and other background noise, leading to a noisy phase signal. Thus, the SNR of a phase-discriminating Φ-OTDR will be deteriorated and it is actually related to both the SR and the length of the vibration region. The RBS trace of Φ-OTDR can be described by the classical one-dimensional impulse response model [17]. The optical power *P*(*t*) of RBS light is given as:(1)$$P\left(t\right)=2\sum_{i}^{N}\sum_{j>i}^{N}a_{i}a_{j}cos\varphi_{ij}exp\left[-\alpha \frac{c\left(\tau_{i}+\tau_{j}\right)}{n}\right]\cdot rect\left(\frac{t-\tau_{i}}{W}\right)rect\left(\frac{t-\tau_{j}}{W}\right)$$
(2)$$rect\left(\frac{t-\tau_{i}}{W}\right)=\{\begin{array}{cc} 1 & if\;0\leq \frac{t-\tau_{i}}{W}\leq 1\\ 0 & else \end{array}$$
where *a_i_* is the amplitude of *i*-th scattered light, *N* is the total number of scattering points within one probe pulse, *α* is the fiber attenuation constant, *c* is the velocity of light in vacuum, *n* is the refractive index of fiber, *τ_i_* corresponds to the time delay for the probe pulse travelling through the distance *L_i_* from the input end to the *i*-th scattering point, *W* is the pulse width in the sensing fiber and *φ_ij_* denotes the phase difference between the *i*-th and *j*-th scattering point, which can be given as:(3)$$\varphi_{ij}=4\pi nL_{ij}/\lambda$$
where *L_ij_* = *L_j_* − *L_i_*, *λ* is the wavelength of laser. When the vibration takes place between the *i*-th and *j*-th scattering point, the length of *L_ij_* will be changed by Δ*L*. Any fiber length variation will lead to a phase difference change Δ*φ*, which can be written as [2]:(4)Δφ=4πnΔL/λ

It is clear that the fiber length variation induced by vibration can be obtained by demodulating the phase difference change Δ*φ*. To simplify the analysis, it is reasonable to assume that the change of fiber length is only caused by an external vibration event and background noise within the SR cell. As we have discussed in the previous section, the background noise level for in-field applications is commonly much higher than the system noise. Therefore, the effect of system noise on SNR was ignored. The fiber length variation between adjacent scattering points induced by vibration can be considered as Δ*l_s_*. Considering that the noise is stochastic, the equivalent fiber length variation Δ*l_n_* between adjacent scattering points caused by background noise can be expressed as:(5)$$\Delta l_{n}^{2}=\frac{1}{N}\sum_{i=1}^{N}\Delta l_{ni}^{2}$$
where Δ*l_ni_* is the fiber length variation of the *i*-th scattering point caused by background noise and *N* is the total number of scattering points within one probe pulse. Thus, the SNR of phase-discriminating Φ-OTDR can be given as:(6)$$SNR\propto \Delta L_{s}/\Delta L_{n}$$
where Δ*L_s_* and Δ*L_n_* are the fiber lengths variation induced by vibration and background noise, respectively. Assuming that the scattering point density per unit fiber length is *Κ* and the length of vibration region is *L*. When SR ≤ *L*, Δ*L_s_* and Δ*L_n_* can be defined as:(7)$$\Delta L_{s}=\Delta l_{s}K\cdot SR$$
(8)$$\Delta L_{n}^{2}=\sum_{i=1}^{K\cdot SR}\Delta l_{ni}^{2}=\Delta l_{n}^{2}K\cdot SR$$

Substituting Equations (7) and (8) into Equation (6), the SNR can be written as:(9)$$SNR\propto \Delta L_{s}/\Delta L_{n}=\frac{\Delta l_{s}}{\Delta l_{n}}\sqrt{K\cdot SR}$$

Clearly, the SNR increases monotonically with SR when SR < *L*. When SR = *L*, we can get the optimum SNR. On the other hand, when SR > *L*, Δ*L_s_* and Δ*L_n_* can be defined as:(10)$$\Delta L_{s}=\Delta l_{s}KL$$
(11)$$\Delta L_{n}^{2}=\sum_{i=1}^{K\cdot SR}\Delta l_{ni}^{2}=\Delta l_{n}^{2}K\cdot SR$$

Substituting Equations (10) and (11) into Equation (6), we obtain the SNR when SR > *L*,
(12)$$SNR\propto \Delta L_{s}/\Delta L_{n}=\frac{\Delta l_{s}}{\Delta l_{n}}\sqrt{\frac{L^{2}}{K\cdot SR}}$$

Equation (12) implies that the SNR decreases monotonically with SR when SR > *L*. Above all, the SNR of a phase-discriminating Φ-OTDR sensing system can be written as:(13)$$SNR\propto \{\begin{array}{ll} \sqrt{SR}, & SR\leq L\\ \sqrt{\frac{L^{2}}{SR}}, & SR>L \end{array}$$

Therefore, when SR equals the length of vibration region *L*, the optimum SNR of a sensing system can be achieved.

To further verify the relationship between SNR and SR, an experiment of phase-discriminating Φ-OTDR based on self-heterodyne detection was proposed. The structure was same as that in Reference [18]. A cylindrical piezoelectric transducer (PZT) was used to generate the vibration event. Pure sinusoidal wave driving signals of 1 kHz were applied on the PZT. The length of fiber wrapped on the PZT was 50 m, corresponding to the length of vibration region was 50 m. The SR of this scheme was dominated by the pulse width *W* in the sensing fiber and the interval *L* between dual pulses, which can be describes as:(14)$$SR=\frac{W+L}{2}$$

The pulse width *W* in the sensing fiber was fixed to 20 m. By changing the interval *L*, SRs from 20 m to 80 m with the tuning step of 10 m could be obtained. After demodulating the phase difference change Δ*φ*, fast Fourier transform (FFT) was used for getting the frequency spectrum of vibration event under a different SR condition. The SNR was considered as the power difference between the peak and the far end noise floor of the frequency spectrum. The blue dots were the experimental data. Equations (9) and (12) were used to fit these experimental data, after which the red fitting curve could be obtained. The monotonicity of the measured SNR with SR, as shown in Figure 1, was consistent with the theoretical analysis.

## 3. Experimental Setup

Based on the above analysis, the underlying qualitative relationship between SR, SNR and the length of vibration region has been revealed. However, it is still difficult to achieve performance optimization through a fixed SR, as the length of a vibration event region cannot be known in advance and multiple vibration events of different vibration regions may occur along the sensing fiber at the same time. The sensing system must have the capability of MSR analysis to obtain performance optimization for the phase-discriminating Φ-OTDR sensing system. Therefore, we presented a novel MSR Φ-OTDR scheme based on FDM, shown in Figure 2.

A narrow linewidth laser was used as the light source, whose linewidth and center wavelength were 3.7 kHz and 1550.12 nm, respectively. The output of laser was separated into three paths by a 3 × 3 optical coupler (OC_1_). Three acoustic optical modulators (AOM_1,2,3_) were used to generate optical pulse with frequency shifts Δ*f* of 40 MHz, 80 MHz and 150 MHz relative to the laser, where AOMs were modulated by a pulse generator. The pulse width and repetition rate of three probe pulses were 100 ns and 25 kHz. These probe pulses were then combined by OC_2_ and amplified by an erbium doped fiber amplifier (EDFA). The optical filter of 0.8 nm bandwidth was inserted to minimize the broadband amplifier spontaneous emission (ASE) noise. The peak power of the probe pulse was amplified to 20 dBm and then launched into 1.62 km sensing fiber through a circulator. Three PZTs were installed on the sensing fiber and the lengths of vibration regions were 30 m, 40 m and 20 m, respectively. Pure sinusoidal wave driving signals of 1 kHz were applied on all three PZTs. The RBS was received by a 200 MHz photodetector (PD). A total of 500 consecutive Φ-OTDR traces were recorded by data acquisition (DAQ) for signal processing with the sampling rate of 1 GSa/s. Figure 3 shows the time domain waveforms of the tri-pulses. The time delays between P_1_-P_2_ and P_2_-P_3_ were 100 ns and 200 ns, by adjusting the triggering time of pulses. Thus, the time delay between P_1_ and P_3_ could be considered as 300 ns. The corresponding intervals of every two pulses were 20 m, 40 m and 60 m, respectively. The on-time of pulse itself was always set to be 100 ns, corresponding to 20 m pulse width *W* in the sensing fiber.

## 4. Results and Discussion

The output of PD will contain the direct current (DC) terms and three product terms of the electric field of the RBS generated from tri-pulses. The time domain waveform and frequency spectrum of one RBS trace are depicted in Figure 4. The frequencies of three beat signals were 40 MHz, 70 MHz and 110 MHz.

Three digital band pass filters with center frequencies of 40 MHz, 70 MHz and 110 MHz were designed to extract these three beat signals. The passband of these three filters was 20 MHz. The time domain waveforms of three beat signals are shown in Figure 5. In this scheme, the probe pulse consisted of tri-pulses with different time delays and every two pulses can generate one beat signal. According to Equation (14), different time delays can result in different SRs, so each beat signal corresponds to one SR. That means, for 40 MHz beat signal, the SR is 30 m, for 70 MHz beat signal, the SR is 40 m, and for 110 MHz beat signal, the SR is 20 m.

The envelope and phase angle of the three beat signals can be extracted by the Hilbert transform [19]. Due to the difference in the order of designed band pass filters and the delay between pulses, there exists delay in each beat signal. This delay can be corrected according to the reflection peak generated at the incident end of the sensing fiber. The waterfall plots of aligned phase angle for 110 MHz, 70 MHz and 40 MHz beating signals are given in Figure 6. Taking PZT1 for example, the regions marked in red rectangular frames show the periodical phase change induced by PZT1. It is clear that the vibration region of 40 MHz was of the highest contrast ratio, implying that the best SNR has been obtained.

The power spectra of demodulated vibration signals at PZT1 and the system noise are shown in Figure 7. The frequency of the first peak was 1 kHz, which was coincident with the frequency of waveform applied on PZT1. By processing three beating signals, the SNRs of PZT1 under different SRs were 47.73 dB, 48.49 dB and 53.28 dB, respectively. Obviously, the red spectrum had the optimum value of SNR, which was obtained by processing 40 MHz beat signal. The length of fiber wrapped on the PZT1 was just right (30 m), so the SNR can reach the optimum value when the SR (30 m) equals the length of vibration region (PZT1). In this experiment, we placed the sensing fiber on a platform without vibration isolation first. Since the laboratory was on the top floor of a six-floor building, the free vibration of the building would directly transmit to the sensing fiber and the PZTs, resulting in relatively strong background noise. Then, in order to evaluate the system noise level, the sensing fiber was put on a rubber vibration isolation pad, which was placed on the floating optical platform to be isolated from the environment, so that it would not be affected by the background noise. For PZT1, the power density of the background noise *P*_bn_ and system noise *P*_sn_ is illustrated in the Figure 7. It is clear that the background noise level was much higher than the system noise level, and the difference in noise level was about 20 dB.

Then, the same signal processing was adopted to the other two vibration regions. The overall measured SNRs of three PZTs are summarized in Table 1. Thus, by MSR analysis, the optimum SNR can be selected and effectively optimize the performance of phase-discriminating Φ-OTDR system.

The method of signal superposition on the digital domain can also increase the SNR by adjusting the equivalent SR [7]. Therefore, a comparison experiment of the advantage of these two SNR improvement methods has been demonstrated. In our experiment, the SR of 70 MHz beat signal was 40 m, which equaled to the length of fiber wrapped on the PZT2. The SR of 110 MHz beat signal was 20 m, which could be considered a finer SR for signal processing the vibration caused by PZT2. As shown in Table 1, for PZT2, the SNR of the signal processed by 40 m SR was about 52.52 dB, whereas when the SR changed to 20 m and without digitally combination, the SNR was 46.67 dB. By digital combination of finer SR, such as moving superposition of two adjacent columns of data, the equivalent SR was reduced to 40 m and the SNR was improved to 52 dB, which was almost the same as the SNR obtained when the spatial resolution matched the length of the vibration region. These results verify the validity of our theoretical analysis that the performance of phase-discriminating Φ-OTDR can reach the optimum when its SR matches the length of the vibration region.

However, as described in Equation (4), in order to get the phase difference change, the upstream and downstream locations of the vibration were selected as reference regions; both reference regions should remain unaffected by the vibration. By demodulating the phase difference between these two reference regions, the vibration can be restored. Using a finer spatial resolution to measure the signal, the reference regions will be still in the vibration region and be influenced by the vibration. When the background noise is extremely strong, the digital combination of finer SR will not further improve the SNR and even may deteriorate the SNR of demodulated phase signal [2]. In any case, the digital combination of signals can simplify the system structure, thus it is also a powerful MSR solution to enhance the system SNR.

In our analysis, the background noise was considered to be induced by the vibration of building. When the sensing fiber is not placed on the optical platform, the background noise should transmit to the sensing fiber without much attenuation. Commonly, the intensity of vibration for a building itself should decrease with frequency. Then, the background noise level of the low frequency band should be higher than that of the high frequency band. In Figure 7, the power density of the background noise level was a little higher when the frequency band was below 5 kHz. However, when the frequency was higher than 5 kHz, the background noise was relatively flat. Therefore, the vibration of the building may not be the only reason behind the background noise. There are several continuous working machines inside the building, such as ventilation systems and central air conditioning systems which can also produce background noise with higher frequency.

The above experimental results have verified that when the SR of Φ-OTDR is consistent with the length of vibration region, the SNR of the sensing system is optimal. The SR and the length of vibration region can be matched by MSR analysis based on FDM as well as digital combination. Of course, the vibration analyzed in the experiment is a sustained single-tone signal and the length of vibration region is exactly equal to the SR of the system, which could not be as easily satisfied in most practical situations. The vibration measured in engineering applications are often not single-frequency signals, and the length of vibration region cannot be predicted in advance. How to evaluate SNR to choose or adjust the SR of the system remains to be a technical difficulty to overcome. However, in the proposed MSR analysis scheme based on FDM, the SR is determined by the time delay of the modulator drive signal, which can be dynamically adjusted through program control. In future work, the length of the vibration region can first be estimated by the amplitude variation of RBS obtained under high SR, and then the SR of the system can be adjusted to make it close to the estimated vibration length, which can achieve the adaptation of the MSR analysis method and improve the overall performance of the sensing system.

## 5. Conclusions

Theoretical analysis and experiments have proven that the optimum value of SNR can be obtained when the SR equals the length of vibration region. Based on this principle, a tri-pulses frequency division multiplexing Φ-OTDR system with multi-spatial resolution analysis property has been demonstrated in this paper. The proposed MSR scheme offers a promising solution for the performance optimization of Φ-OTDR systems that can restore simultaneous vibration events of different vibration region lengths with optimum SNR in one single measurement, and without reducing the detection frequency range.

## Figures and Tables

**Figure 1 sensors-19-00083-f001:**
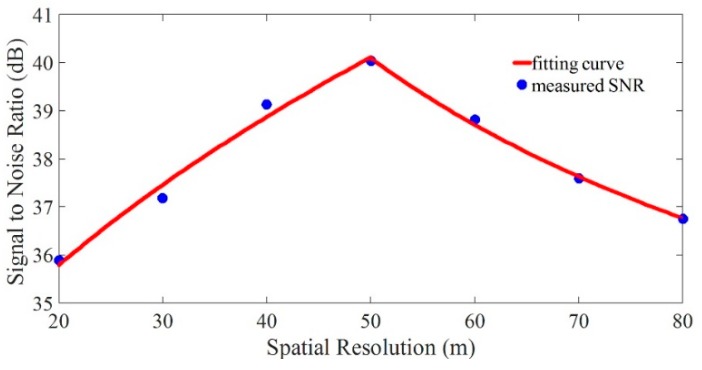
The curve of measured signal-to-noise ratio (SNR) changed with spatial resolution (SR).

**Figure 2 sensors-19-00083-f002:**
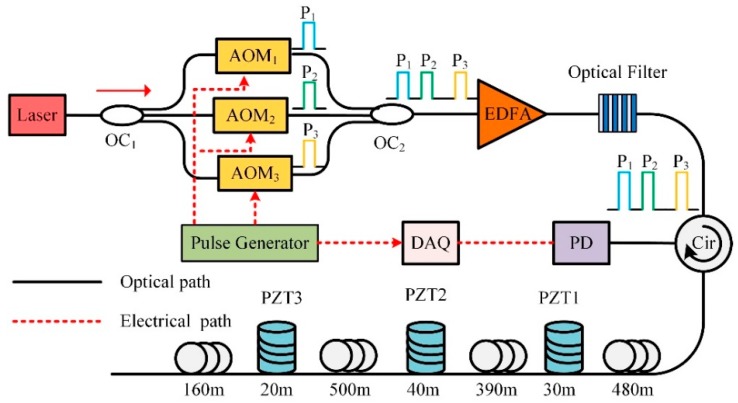
The scheme of the proposed phase-sensitive optical time domain reflectometry (Φ-OTDR) sensing system.

**Figure 3 sensors-19-00083-f003:**
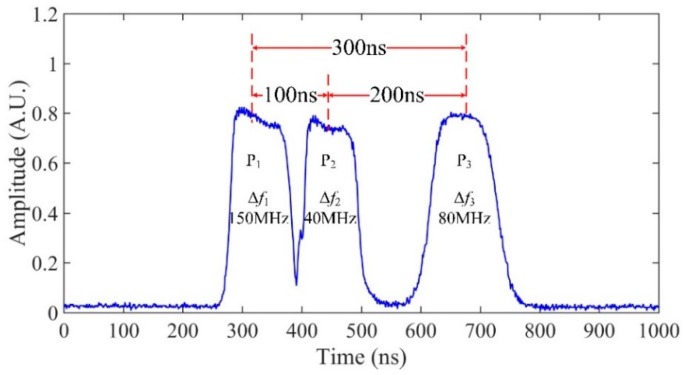
The time domain of three probe pulses.

**Figure 4 sensors-19-00083-f004:**
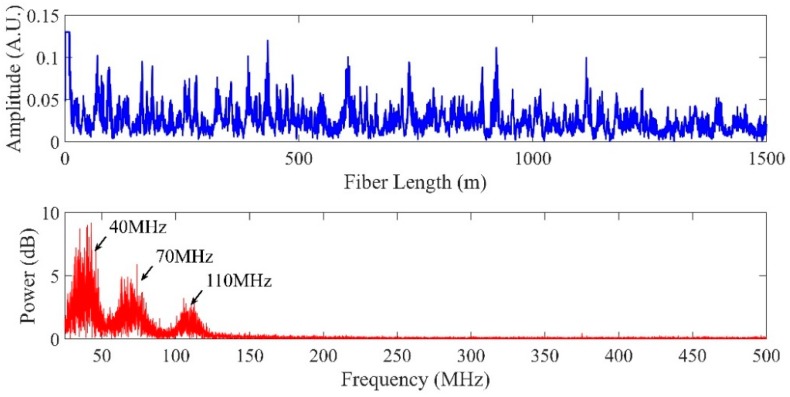
The time domain waveform and frequency spectrum of the output signal of the photodetector (PD).

**Figure 5 sensors-19-00083-f005:**
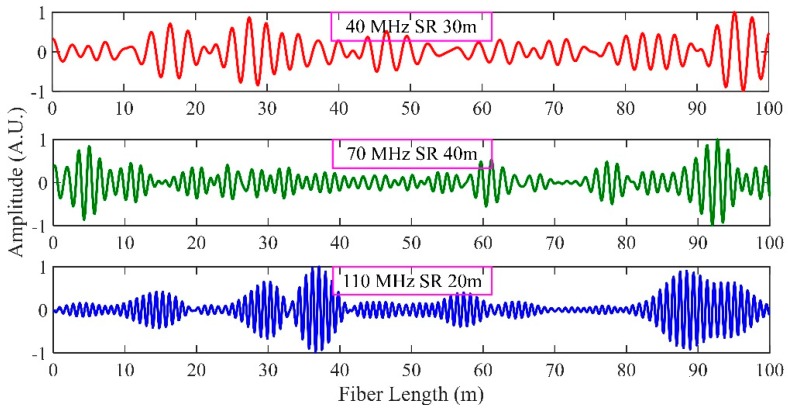
The time domain of three beat signals.

**Figure 6 sensors-19-00083-f006:**
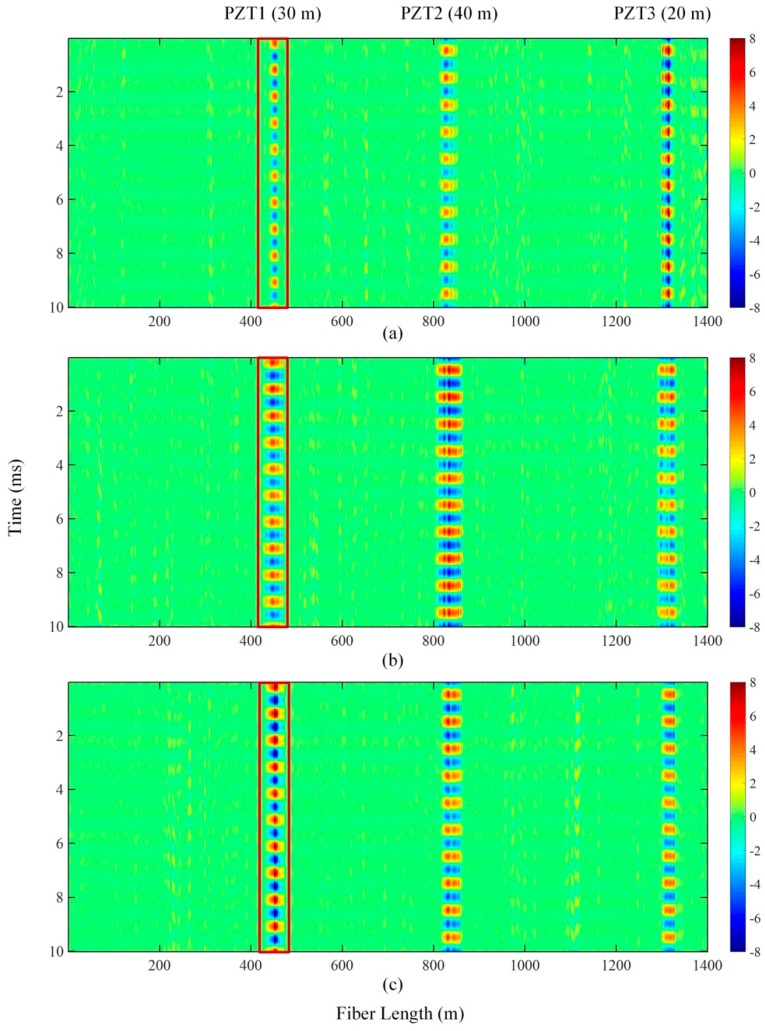
The waterfall plot of Φ-OTDR phase angle for three beat signals. (**a**) 110 MHz; (**b**) 70 MHz; (**c**) 40 MHz.

**Figure 7 sensors-19-00083-f007:**
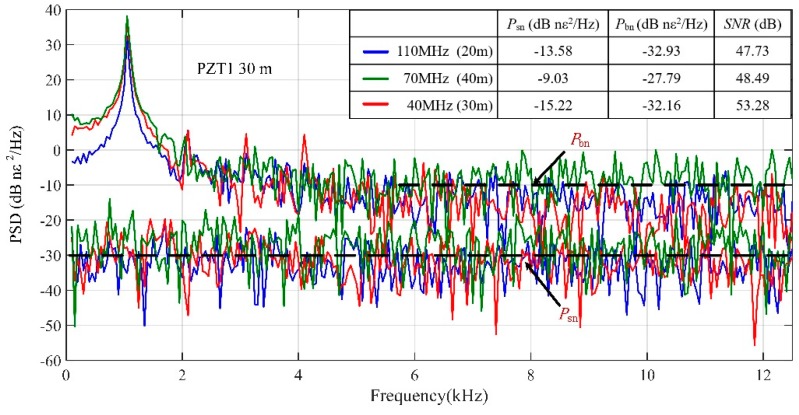
The power spectra of demodulated phase difference change signal and background noise at piezoelectric transducer 1 (PZT1).

**Table 1 sensors-19-00083-t001:** The SNRs of three PZTs at different SRs (unit is dB).

	SR (m)	20	30	40
PZT (m)				
20		**50.46**	47.59	46.66
30		47.73	**53.28**	48.49
40		46.67	48.71	**52.52**

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
