# Peer review of "Performance Optimization for Phase-Sensitive OTDR Sensing System Based on Multi-Spatial Resolution Analysis"

_sensors, 2018, doi:10.3390/s19010083_

Round 1
Reviewer 1 Report
This work investigates the relationship between spatial resolution, vibration region, and SNR in OTDR. They introduce a frequency multiplexing approach to simultaneously record OTDR traces with multiple spatial resolution sizes and show that matching the sensor spatial resolution to the size of the vibration region provides the optimal SNR. This approach is potentially interesting, but I have several comments that should be addressed:
1) The authors compare the SNR of a single measurement with spatial resolution matched to the size of the vibration region to a single measurement with a smaller spatial resolution. However, if the signal was measured using finer spatial resolution, the signal from different spatial resolution cells could be combined digitally to provide a higher SNR. Thus, I think a more fair comparison of the advantage of matching the spatial resolution to the size of the vibration cell would account for the array gain available to the higher resolution measurement.
2) Recording multiple interference signals on the detector simultaneously will degrade the SNR with which you measure each individual interference signal due to increased shot noise. The authors should discuss/analyze the impact of simultaneously recording signals from multiple spatial resolutions on a single detector.
3) At a given time, the beat signals recorded on the detector from different pairs of pulses will come from different positions in the fiber since the pulses had varying delays entering the fiber. I could not find where the authors accounted for this. That is, In Fig. 4 and 5, how did the authors convert time on the detector to position in the fiber, and was this conversion different in Fig. 5 for the different beat signals? In Fig 6—is this why the PZT positions appear to be different in the 110 MHz and 40 MHz waterfalls?
4) Have the authors calibrated the strain introduced on the PZT? In order to compare the performance of this sensor with other OTDR sensors, I would suggest reporting the minimum detectable strain PSD in strain^2/Hz.
5) In Fig. 1, is the fitting curve obtained analytically from equation 12? What value for delta_l_n was used? Did you calibrate the strain introduced by the PZT to get a value of delta_l_s to obtain the fit?
6) I have some questions about the SNR expression in Eq. 6. This does not include the influence from noise sources such as shot noise, laser phase noise, or detector noise that can often limit the performance of an OTDR system. Should this expression be considered a limit on the highest possible SNR you could achieve with this phi-otdr scheme? The authors should discuss the origin of the fiber length variations due to background noise in more detail. How did you calculate delta_L_n?
7) How much power did the pulses have going into the fiber/after the EDFA?
Author Response
Please find authors' response in the attachment.
Reviewer 2 Report
I support the publiction of the paper subject to minor corrections/improvements:
Line 35: RBS definition?
Line 57 I suggest: Many advanced DSP techniques to improve the sensing performance of phase-discriminating Φ-OTDR have been reported in literature.
Line 62&64 I suggest: relationship, and in ther places I would use singular
Line 127: font 'as'
Fig.2 and text describing it: it is not clear how 100 ns/25kHz pulses are generated. AOM produces a frequency shift. What makes the pulses?
Line 156: amplified not 'gained'
Line 190: should be 'frequency spectra'
Line 211: 'situations'
Line 214: 'a technical difficulty to overcome'
Author Response

(The authors gave the same response as above.)

Round 2
Reviewer 1 Report
The authors addressed most of my concerns. I have two remaining comments:
(1) In response to point 1, the authors stated:
“In our experiment, the spatial resolution of 70 MHz beat signal is 40m, which equals to the length of fibre wrapped on the PZT2 is 40m. The spatial resolution of 110 MHz beat signal is 20m, which can be consider as a finer spatial resolution for signal processing the vibration caused by PZT2. The frequency spectrums are shown in the Figure1. When the spatial resolution is 40 m, the SNR is about 52.52 dB. When the spatial resolution is 20 m and after digitally combination, the SNR is 52 dB, which is still lower than the SNR obtained when the spatial resolution matches the length of vibration region.”
The above analysis gets to the point of my comment: that a more fair comparison of the 20 m sensor would, in this case, provide SNR within 0.5 dB of the 40 m sensor. However, I don’t think this updated analysis was included in the revised manuscript. I would suggest adding this to the manuscript since it would be easy to digitally compute the response for different sensor sizes from a higher resolution measurement and recover most of the SNR gains possible by matching the sensor size to the size of the vibration event.
(2) Showing Fig. 7 as a PSD is definitely helpful. However, I am a little surprised that the environmental background noise is flat out to 12 kHz. Typically, I would expect environmental/background noise to show more spectral features and to generally roll off with frequency. Similarly, the “System noise” is perfectly flat all the way to DC. In this case, is there really no environmental noise pick up at low frequencies? Is the “optical platform” just a floating optical table?
Author Response
Please see authors' response in the attachment.
